# Biological Activity of the Carrier as a Factor in Immunogen Design for Haptens

**DOI:** 10.3390/molecules23112977

**Published:** 2018-11-14

**Authors:** Özlem Ertekin, Esin Akçael, Harun Kocaağa, Selma Öztürk

**Affiliations:** TÜBİTAK, The Scientific and Technological Research Council of Turkey, Marmara Research Center, Genetic Engineering and Biotechnology Institute, 41470 Gebze, Kocaeli, Turkey; esin.akcael@tubitak.gov.tr (E.A.); harun.kocaaga@tubitak.gov.tr (H.K.); selma.ozturk@tubitak.gov.tr (S.Ö.)

**Keywords:** carrier protein, hapten immunization, transferrin receptor, immunogen, endocytosis

## Abstract

Immunoanalytical methods are frequently employed in the detection of hazardous small molecular weight compounds. However, antibody development for these molecules is a challenge, because they are haptens and cannot induce a humoral immune response in experimental animals. Immunogenic forms of haptens are usually prepared by conjugating them to a protein carrier which serves as an immune stimulator. However, the carrier is usually considered merely as a bulk mass, and its biological activity is ignored. Here, we induced an endocytic receptor, transferrin receptor, by selecting its ligand as a carrier protein to enhance antibody production. We conjugated aflatoxin, a potent carcinogenic food contaminant, to transferrin and evaluated its potential to stimulate antibody production with respect to ovalbumin conjugates. Transferrin conjugates induced aflatoxin-specific immune responses in the second immunization, while ovalbumin conjugates reached similar antibody titers after 5 injections. Monoclonal antibodies were successfully developed with mice immunized with either of the conjugates.

## 1. Introduction

An antigen is any molecule that can bind to the surface receptors of immune system cells and antibodies. An immunogen, on the other hand, is a molecule which not only binds to these molecules, but is also capable of triggering the cell-mediated immune response or humoral immune response and the production of antibodies [1,2,3]. All immunogens are at the same time antigens, but the reverse is not true. There are non-immunogenic antigens, like some bacterial toxins which non-specifically induce the immune system to exhaustion which are called “superantigens” [1], or some molecules which are too small to be immunogenic are called “haptens” [4,5]. Haptens are not immunogenic because even though they may pose a threat to the organism, like in the case of mycotoxins, they can be detoxified by chemical modification with liver enzymes, and there is no need to start the energy consuming [6] humoral immune response. But we sometimes need a humoral immune response against hapten molecules, because we need antibodies to use in immunodiagnostic systems [7,8]. To be able to induce such immunogenicity in these non-immunogenic antigens, we first need to understand the mechanism of antibody production.

Antibodies are produced as the defense molecules of the body through humoral immune response. B cells of the immune system are specialized cells for the production of antibodies. In the mammalian body, there is a repertoire of mature B cells, each of which express a different cell surface receptor. These receptors are immunoglobulins embedded the cell membrane which serve as B-cell receptors (BCR). Due to extensive genetic variability, which is beyond the scope of this paper, there may be more than 10^11^ different receptors with different antigen specificities [4]. A mature B cell lives for 3–4 days and dies unless it encounters the specific antigen with structural compatibility to its BCR. The antigen encounter is necessary but not enough for antibody production. The B cell needs the assistance of another group of immune cells called T cells for stimulation, and T cells also need to be activated via antigen encounter. However, the encounter of T cells with the antigen should be through specialized molecules called “major histocompatibility complexes” (MHCs) on the surface of antigen presenting cells (APC). When an antigen enters the body, it is internalized by a group of cells called APCs, which include dendritic cells, macrophages, and B cells themselves. The internalization is either by phagocytosis or receptor mediated endocytosis. After internalization, the antigen is processed by digestion into fragments, and combined with MHC molecules to be presented on the cell surface [9]. Only peptides can be presented via MHC molecules; therefore, proteins are the strongest immunogens, and haptens cannot elicit immune response. In order to induce the immunogenicity of haptens, whether they are needed for antibody development [10,11] or vaccine studies [12,13], they are chemically conjugated to immunogenic proteins, and act as a part of that protein.

So, the cascade of antibody production starts with the internalization of the antigens by APC; this fact can be used to develop a strategy to raise a better immune response against haptens. For instance, cationized proteins are established as superior carriers, since their positive charge makes phagocytosis easier due to the negative charge of the cell membrane [14]. In this work, we aimed to increase the immunogenicity of the hapten by inducing receptor-mediated endocytosis via transferrin receptors (TFR) on the cells.

Transferrin (TF) is a vital protein facilitating iron uptake by the cells. Iron bound TF (holo-TF) has high affinity to TFR at physiological pH. Circulating TF binds to TFR when it is loaded with iron and internalized by receptor mediated endocytosis. Acidic pH of the endosomic lumen causes a conformational change in TF, which causes the release of iron. Iron-free TF (apo-TF) has an affinity to TFR at this pH. Once the TFR is recycled and returns to the cell membrane, apo-TF is released to the extracellular matrix [15,16]. Since this cycle is vital for the cell, TFR is used as a proliferation marker for the cells where highly proliferating cells express more TFR, and serves as a target for therapy. Blocking of TFR with monoclonal antibodies poses a cytotoxic effect, and this is proposed as a mechanism to inhibit cell growth [17,18]. Another strategy utilizing the TFR is using it for the targeted delivery of drugs in malignancies, since highly-proliferating cells express more TFR than their healthy counterparts [19,20]. It was also shown that the binding of peptides or nucleic acid ligands other than TF can also induce receptor-mediated endocytosis via TFR [21,22]. Previous work has established the intersection of the TF endocytic pathway with MHC2 exocytic route, and it was demonstrated that antigens endocytosed by TFR can be presented by MHC2 molecules [23].

Here, we proposed the use of TF as a carrier molecule in hapten immunizations for enhanced antibody production. We used aflatoxin (AF) as the model hapten in this study. AF is a hepatotoxic mycotoxin classified as class 1 human carcinogen by International Agency for Research on Cancer, and antibodies against AF are widely used in its detection and quantification [24]. TF-AFB1 and ovalbumin (OVA)-AF conjugates were comparatively evaluated in terms of induction of AF-specific antibody response in BALB/C mice. A cross species TF was required to eliminate the immunogenic disadvantages related with self-tolerance. The protein sequence blast resulted in 73% identity with the closest non-rodent TF ortholog [25]. When the protein sequences are aligned by Needle program, murine TF (mTF) and human TF (hTF) has a 72.7% sequence identity and 82.8% sequence similarity. The domains of TF responsible for TFR binding were previously identified with structural studies. It was shown that aminoacids at positions 71–74 and 142–145 were involved in TF-TFR interaction, and that scattered aminoacids between positions 349–372 were also shown to play role [26]. When we compared the aminoacid sequences of the mTF and hTF at the well-defined areas 71–74 and 142–145, we observed 80% identity and 100% similarity. So, hTF was selected as the cross-species immunogenic carrier.

## 2. Results

AFB1, the most potent AF derivative, was used as the model hapten in the immunization work. AFB1 was conjugated to hTF and OVA for mice immunizations. Since the mice immune system is responsive to non-AF epitopes of the prepared immunogen, we expected the mice sera to have antibodies against both the carrier and AFB1. So, in order to be able to measure the AF specific immune response, we conducted the enzyme-linked immunosorbent assay (ELISA) tests using BSA-AFB1 conjugate as a coating agent. We prepared 3 different conjugates of AFB1: BSA-AFB1, OVA-AFB1, and TF-AFB1.

### 2.1. Preparation and Characterization of AF–Protein Conjugates

AFB1-protein conjugation was conducted by modification of the method of Zhou et al. in two steps [27]. In the first step, the proteins were cationized by the enrichment of surface amine groups [28]. The cationized proteins were in turn conjugated to AFB1 by a Mannich type reaction [27]. Wave scans of unconjugated proteins, AFB1 and AFB1-protein conjugates, are presented in Figure 1 for BSA, OVA, and TF conjugation reactions. The maximal absorption wavelength of proteins is at 278 nm and AFB1 is at 366 nm. AFB1 also has a minor peak at 266 nm. The wavescans of the protein conjugates showed two spectrally active regions. The maximal absorptivity of OVA and TF conjugates was 276 nm, and that of BSA-AFB1 was 270 nm. A minor peak at 370 nm was observed in all three conjugates. The presence of two different regions of spectral activity and the blue shift in the major protein peak showed the success of the conjugation. The AF:protein conjugation ratios of the conjugates were calculated as 1.14, 0.95, and 0.81 for BSA, OVA, and TF conjugates, respectively.

### 2.2. Immune Response

We evaluated the immunization efficacy of the TF-AFB1 conjugate with respect to a similarly-prepared OVA-AFB1 conjugate. The efficacy of the immunizations was monitored by testing the sera of BALB/c mice with indirect ELISA. When an animal is immunized with the conjugated form of a hapten, the sera of the animal may contain antibodies against the carrier protein, hapten, or the bond between them. Immune sera obtained from the OVA or TF conjugate-immunized mice were tested using the BSA-AFB1 conjugate to observe only AF-specific antibodies by eliminating the signal resulting from carrier immunogenicity. The results of indirect ELISA showed an earlier AF specific immune response in the mice immunized with TF-AFB1 conjugate when compared with OVA-AFB1 immunized mice. TF-AFB1-immunized mice showed high antibody titers, even in the initial immunizations, while OVA-AF immunized mice showed no AF specific response in the first four immunizations. However, after the 4th immunization, antibody titers induced by both immunogens reached similar levels (Figure 2).

The indirect ELISA showed the hapten induced immune response in the animal. However, the hapten is in a protein-conjugated form, and the conjugate of the hapten may be structurally different from the unconjugated, free hapten. So, there is a possibility that the developed antibodies recognize the AF when conjugated to the protein, but do not interact with free AF in solution. The signal obtained by indirect ELISA might be for AFB1, the chemical bond between AFB1 and protein, or to the protein itself. So, an Indirect Competitive ELISA (IC-ELISA) method was employed in order to confirm the AF specific immune response in TF-AFB1-immunized mice after the 4th immunization (Figure 3). In the IC-ELISA tests, we coated the ELISA plates with four different antigens. The coating antigens, competing antigens, and the expected results of competition is summarized in Table 1. Immune sera were incubated with either free AFB1 or the carrier protein used in immunizations prior to loading on ELISA plates. Incubation without the antigen was used as a negative control. If the antibodies in the sera interact with AFB1 or the carrier protein, they are blocked and cannot bind to the antigens coated to the ELISA plates. So, for IC-ELISA, a lower signal is the evidence of the interaction of the antibody with the competing antigen (Figure 3).

The test was used to show the interaction of the immunized mice sera with soluble AFB1. The results of the IC-ELISA conducted with the 4th immunization sera of highest responding 3 TF-AFB1-immunized mice and the 3 OVA-AFB1-immunized mice are presented in Figure 4. Control sera of TF-AFB1 immunized mice interacted with TF-AFB1, TF, and BSA-AFB1, but did not bind to BSA-coated wells. Pre-incubation with TF decreased the binding of sera to TF-AFB1 and TF coated wells where the signal in TF coated wells decreased 80%. Pre-incubation with AFB1 completely abolished the binding to BSA-AFB1, indicating that all antibodies binding to BSA-AFB1 were also binding to free AFB1. AFB1 competition also decreased the signal in TF-AFB1-coated wells. A similar pattern was observed with OVA-AFB1-immunized mouse serum. However, the response to free AFB1 was lower in this case, since sera from the 4th immunization were used (Figure 4). This result also showed that carrier-specific antibodies were produced earlier than hapten-specific antibodies. 

### 2.3. Monoclonal Antibody Development

The efficacy of the TF as a carrier protein in hapten-specific antibody development was further evaluated by testing the potency of TF-AFB1-immunized mice as lymphocyte donors for antibody development with hybridoma technology. Hybridoma fusions were done using two mice from each immunization set. Mice with the highest serum antibody titers for free AFB1 were selected for monoclonal antibody development work. Four fusions with two TF-AFB1 and two OVA-AFB1 -immunized mice were performed. The results of the fusions were summarized in Table 2. Fusions yielded a total of 3852 hybrid clones, with 685–1540 hybrid clones per fusion. Every cell culture well containing hybrid cells was tested with indirect ELISA using BSA-AFB1-coated plates for antibody production. A total of 134 clones were selected, whose cell culture supernatants interacted with BSA-AFB1. These results are not enough to draw a conclusion for comparison of the two carriers, but it was clearly shown that TF can be used as an effective carrier for hapten-specific monoclonal antibody development.

## 3. Discussion

In this work, we aimed to increase the efficacy of hapten immunizations by using a carrier protein which facilitates antigen presentation by stimulating endocytic pathway. For this aim, we evaluated efficacy TF as a carrier protein to raise the AF specific humoral immune response in BALB/c mice with respect to OVA. We conjugated AF to cationized hTF by a Mannich-type method, where carboxylic acid groups of the positively-charged aminoacids were converted to amine groups. The use of cationized proteins both contributed to increasing the efficacy of conjugation reaction [27] and was an additional enhancing factor which increased the immunogenicity [29]. The absence of positively-charged aminoacids in the well-established TFR binding domains of hTF (142 GRSA 145 and 71 IAAN 74) [26] suggested that receptor binding would not be affected. Our results clearly showed that the use of TF significantly accelerated the hapten-specific humoral immune response in mice.

We evaluated the mice sera at the 4th immunization where TF conjugate-immunized mice was responsive, and OVA conjugate-immunized mice was not responsive, i.e., did not demonstrate that the antibodies in TF conjugate-immunized mice raised earlier than those in the OVA conjugate immunized-mice were hapten-specific. OVA-AFB1-immunized mice neither interacted with BSA-AFB1 coated wells, nor was a significant decrease observed in OVA-AFB1-coated wells upon AFB1 competition. On the other hand, hTF-AFB1-immunized mice sera interacted with BSA-AFB1 coated wells and were also responsive to AF competition in IC ELISA. In hapten immunizations, the late response is known to be an issue in antibody development studies, and usually, at least 4–6 immunizations are required to observe hapten-specific antibodies [27,30], unlike immunogenic proteins, which demonstrate high titers even after second immunization. The use of TF as a hapten carrier actually induced an immunogenic protein-like response in mice, and yielded AF specific antibodies even after the second immunization.

TF has been used as an intracellular drug delivery agent, since every mammalian cell contains TF receptors [19,20,31]. However, when it was used for the delivery of small molecular weight drugs with hapten structures, such as peptides, it was shown that, a strong anti-drug immune response was observed, and drug delivery efficiency decreased [32]. Although similar works demonstrate the use of TF as a drug delivery agent, there are few studies utilizing it as a carrier protein for haptens [33,34,35]. Proteins that are routinely utilized as carriers in hapten immunizations are IgG, OVA, or BSA [36,37]. In this work, we elucidated the efficiency of TF as a carrier protein in AF immunization to obtain earlier immune responses from the experimental animals. Monoclonal antibodies were successfully developed using mice immunized with either of the carrier proteins.

This work introduced the biological activity of the carrier molecule as a critical factor in immunogen preparation for hapten molecules, and opens possibilities for the use of other endocytic receptor ligands as carrier molecules. It was previously shown that TFR can be induced by some peptide sequences and RNA molecules [21,22]. So, these motifs can be introduced to the carriers of vaccines against the hapten (for instance opioid drugs), and facilitate the development of more efficient and safer vaccines. Recently, some work has focused on different TFR1 and TFR2 and their differential substrate specificity [16]. So, while different receptors of the same species may have differential ligand specifities, it must be noted that a careful selection of TFR-binding motifs or carriers is required, since TFR of different species may have differential binding patterns.

## 4. Materials and Methods 

### 4.1. Materials and Equipment

Most chemical reagents, such as buffers and salts, were purchased from Sigma Aldrich (St. Louis, MO, USA), with the exception of 1-Ethyl-3-(3-dimethylaminopropyl) carbodiimide (EDC) (Thermo Scientific, Waltham, MA, USA) and AFB1 (Fermentek, Jerusalem, Israel).

ELISA readings were done with a Biotek Synergy HT microtiter plate reader, which was controlled by a personal computer containing the GEN5 standard software package from BioTek Instruments, (Winooski, VT, USA). Biochrom Libra S2 spectrophotometer was used in spectrophotometric measurements.

### 4.2. Preparation of AF–Protein Conjugates

Carboxyl groups of the proteins were converted to amine groups using the method of Domen, 1992 [28], where ethylene diamine (EDA) was conjugated to the carboxyl with an EDC linker. To this end, 670 µL EDA was mixed with 5 mL 2-(*N*-morpholino)ethanesulfonic acid (MES) buffer (0.1 M, pH: 4.8), and 25 mg of proteins were dissolved in EDA solution. The reaction was catalyzed with 18 mg of EDC. The reaction was stopped with the addition of 150 µL sodium acetate (4 M, pH: 4.8) after 1 h. AFB1-protein conjugation was achieved by Mannich type reaction [27]. Cationized BSA, OVA, or TF was conjugated to 40-fold molar excess of AFB1 (2 mg/mL in dimethylformamide) in 0.1 M MES, pH: 4.8 in the presence of formaldehyde (40-fold excess with respect to AFB1). Unconjugated AF and other reaction components were cleaned by exhaustive dialysis against 0.1 M MES, pH: 4.8. The hapten density of the prepared conjugates were calculated according to the following formula derived from Beer-Lambert Law: C(AF)/C(BSA) = Abs_360_ × εBSA_280_ /(Abs_280_ × εAFB2_360_ − εAFB2_280_ × Abs_360_) where A: absorbance, ε: molar extinction coefficient, C: concentration [27].

### 4.3. Immunizations

Six-to-eight weeks old, female Balb/c mice were intraperitoneally immunized with TF-AFB1 or OVA-AFB1 (5 mice/immunogen) in phosphate buffered saline (PBS: 10 mM K_2_HPO_4_, 10 mM KH_2_PO_4_, 0.15 M/L NaCl, pH 7.2). Initial immunization was done with 20 µg antigen with complete Freund’s adjuvant. Four booster immunizations were done with 50 µg immunogen with incomplete Freund’s adjuvant. The second immunization was one week after the initial immunization, and subsequent injections were with 2-week intervals. Mice sera were collected 10 days after each immunization, starting from the second immunization. Two mice with the highest antibody titers for each immunogen were selected for monoclonal antibody development, and received an intravenous booster immunization. TF-AFB1 immunized mice received 50 µg OVA-AFB1 booster and vice versa in order to enrich only AF specific B cells. All animal experiments were approved by the ethical committee of Turkish Scientific and Research Council (TÜBİTAK), Marmara Research Center (MAM), Genetic Engineering and Biotechnology Institute.

### 4.4. Indirect ELISA

Indirect ELISA [38,39] was used to monitor mice immune response and for the screening of hybridoma supernatants. BSA-AFB1 (500 ng) was coated to ELISA plates overnight at 4 °C in PBS. Wells were blocked with skimmed milk (0.1% in PBS). Two thousand-fold PBS diluted mice sera or the hybridoma culture supernatant (100 µL) were incubated for 1 h at 37 °C. Visualization was done by goat anti-mouse polyvalent (IgA, IgM, IgG) antibodies (Sigma A0162) labeled with alkaline phosphatase as secondary antibody and p-nitrophenyl phosphate (PNPP, Sigma A4744) (1 mg/mL in substrate buffer: 1 mM ZnCl_2_, 1mM MgCl_2_, 0.1 M glycine, pH 10.4). Optical density at 405 nm was recorded with a multiplate reader (Biotek Synergy HT, Winooski, VT, USA).

### 4.5. Indirect Competitive ELISA

AF-specific immune response in mice sera was confirmed by testing the interaction of antibodies with soluble AF with IC-ELISA [40]. We also used IC-ELISA to observe the carrier-specific antibodies in mice sera. ELISA plates were coated with 500 ng AF-protein conjugate or 100 ng unconjugated protein in PBS and blocked, as described previously. Soluble AFB1 (1 µg/µL serum) or proteins (200 ng/µL serum) were incubated with 5000-fold diluted mice sera for 30 min at 37 °C. Then, 100 µL of the preincubation mix was transferred to the coated ELISA plates and incubated at 37 °C for one hour. Visualization of the bound antibodies was achieved as explained in the indirect ELISA method.

### 4.6. Development of Monoclonal Antibodies

Monoclonal antibodies were developed with the hybridoma technique [41]. The spleen and lymph nodes of selected mice were used as lymphocyte sources in the fusion studies. The lymphocytes of the immunized BALB/c mouse and mouse myeloma cells (F0; ATTC CRL 1646 ) were fused in the presence of 50% polyethylene glycol (PEG) 4000 [42,43,44]. The fusion product was resuspended in Dulbecco’s Modified Eagle Medium (DMEM) containing 20% fetal calf serum and antibiotic, and distributed into a peritoneal cell containing 96 well culture plates. The plates were incubated at 37 °C, 5% CO_2_, 95% humidity. At 10–15 days after fusion, plates were screened with a reverse phase light microscope for hybrid development. Supernatants from hybridoma colonies were screened using indirect ELISA with BSA-AFB1 coated plates.

### 4.7. Ethical Considerations

Animal experiments were performed in compliance with the appropriate laws and institutional guidelines with the approval of TÜBITAK Marmara Research Center, Genetic Engineering and Biotechnology Institute ethical committee.

### 4.8. Safety Considerations

AFs are known to be hepatotoxic and AFB1 is a class 1 human carcinogen. Therefore, all experiments were conducted at a controlled Biosafety level 2 laboratory with proper protection.

## Figures and Tables

**Figure 1 molecules-23-02977-f001:**
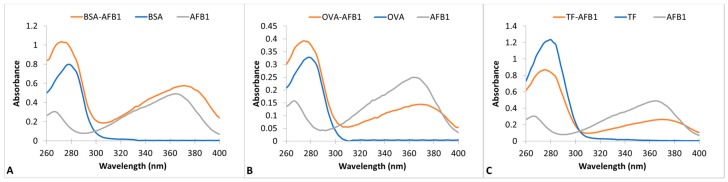
UV-VIS absorption spectra of AFB1-Protein conjugates between 260 nm–400 nm wavelengths. (**A**) BSA-AFB1, BSA, and AFB1; (**B**) OVA- AFB1, OVA, and AFB1; (**C**) TF-AFB1, TF, and AFB1.

**Figure 2 molecules-23-02977-f002:**
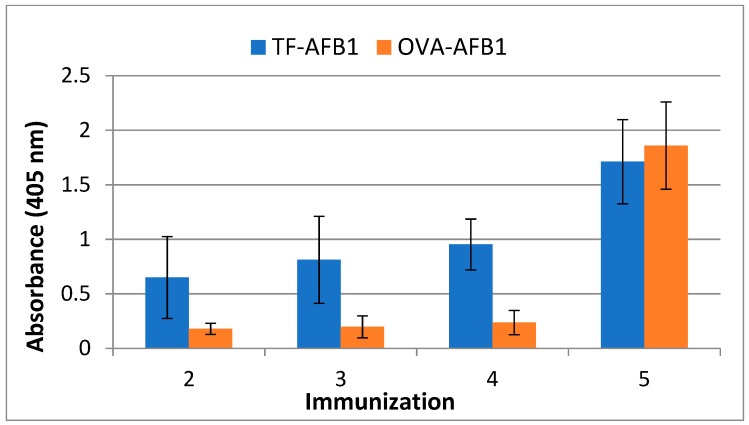
Comparison of AF specific immune response of mice immunized with OVA-AFB1-AFB1 or TF-AFB1. Mice sera were 2000 times diluted with PBS. The results are presented as a mean of antibody titers of 5 mice. Error bars represent standard deviations.

**Figure 3 molecules-23-02977-f003:**
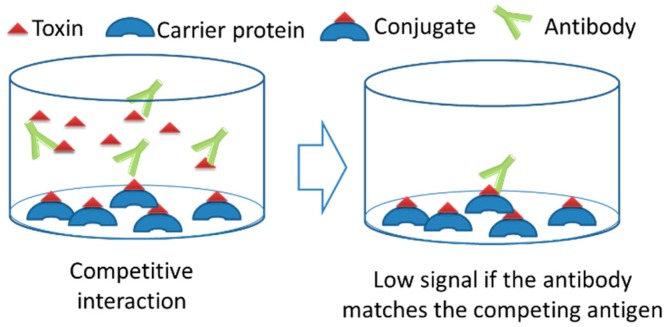
Schematic representation of competitive interaction.

**Figure 4 molecules-23-02977-f004:**
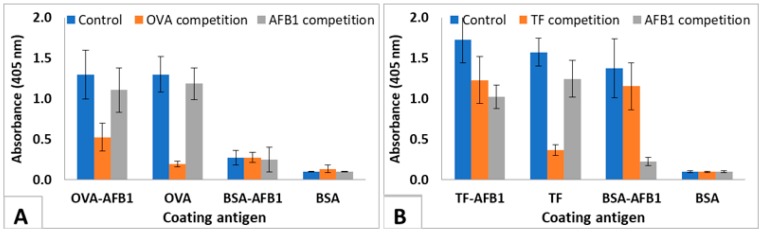
IC-ELISA showing interaction of mice sera with AFB1 and proteins. (**A**) OVA-AFB1-immunized mice sera with AFB1 (200 ng/µL serum) and OVA (1 µg/µL serum). (**B**) TF-AFB1-immunized mice sera with AFB1 (200 ng/µL serum) and TF (1 µg/µL serum). Serum dilutions are 1/5000. The results are presented as mean of 3 replicates. Error bars represent standard errors.

**Table 1 molecules-23-02977-t001:** Interpretation of IC-ELISA results.

Coating Antigen	Detected Antibodies	Competing Antigen	Blocked Antibodies	Signal Change
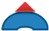	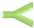	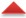 or 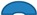		
OVA-AFB1 or TF-AFB1 (conjugates used in immunizations)	AFB1 and carrier protein	AFB1	Anti-AFB1	Partial reduction
Carrier protein	Anti-carrier	Partial reduction
OVA or TF (Unconjugated carrier used in immunizations)	Carrier protein	AFB1	None	No change
Carrier protein	Anti-carrier	No signal
BSA-AFB1 conjugate	AFB1	AFB1	Anti-AFB1	No signal
Carrier protein	None	No change
BSA	Negative control	AFB1	NA *	No signal
Carrier protein	NA *	No signal

* Not Applicable.

**Table 2 molecules-23-02977-t002:** Hybridoma fusion conditions and results.

Fusion	Immunogen	Myeloma Cells	Lymphocytes	Hybrid Cells	Antibody Producing Hybridomas
1	TF-AFB1	0.45 × 10^8^	4 × 10^8^	748	19
2	TF-AFB1	2 × 10^8^	5.92 × 10^8^	878	31
3	OVA-AFB1	2.4 × 10^8^	19.6 × 10^8^	1541	13
4	OVA-AFB1	2.45 × 10^8^	8.28 × 10^8^	685	71

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
