# Peer review of "Biological Activity of the Carrier as a Factor in Immunogen Design for Haptens"

_molecules, 2018, doi:10.3390/molecules23112977_

Round 1
Reviewer 1 Report
In the present manuscript, the authors propose the use of transferrin as a carrier molecule in hapten immunizations for enhanced antibody production in the response to aflatoxin as the model hapten. The efficiency of TF as carrier protein was demonstrated in inducing an earlier immune response in the experimental animals. The response as tested with polyclonal sera was at least partly hapten-specific. Further, antibody-producing hybridoma clones were produced from cells obtain from animals immunized with aflatoxin-transferrin as immunogen, approximately at the same success level as when aflatoxin-OVA was used for immunization. The paper is clearly written, but I would have the following remarks:
- Experimental details should be more concise, such as specifying the reagents for ELISA: how were the polyclonal reagents used for ELISA controlled, what were the buffers used, etc.
- Discussion part is very short, and some points could be raised: even if the authors do not include further characterization of the monoclonal antibodies produced, what would they expect? What are the ways to improve TF-based immunization and make it available for other platforms (such as transgenic rats, llamas, etc.)?
The authors should review the manuscript for the consistency of labels and abbreviations. These would be my minor remarks:
Line 112: UV-VIS in caps
Line 145: Table 1: NA means probably not applicable?
Line 151: completely abolished the binding or similar would be better than completely diminished the signal
Figure 3: Y-axis is not labelled
Table 2: probably the numbers of cells used should be in a form “x10exp8”? Then the same format could be used for all figures shown and there is no need for the reader to recalculate.The table should contain the labels “Fusion 1/Fusion 2” or “Mouse 1/Mouse 2” to be self-explanatory.
Line 180: many studies or similar
Line 195: also the abbreviation EDC should be written out (as EDA); authors should adhere to the same style throughout the text (e.g. also for MES, DMF, etc.)
Line 199: (as well as Figure 3 and throughout the text): authors should consistently use decimal point and not comma
Line 204: the second immunization followed (or similar), please correct the incomplete sentence
Line 218: alkaline phosphatase
Line 226: a full-stop too much. Refer to previous paragraph of your text instead of saying “as previously”
Line 235: CO2, 2 in subscript
Line 244: safety level of the laboratory should be indicated
Author Response
Dear reviewer,
Thank you for your time and careful review of the manuscript. We believe that your thorough evaluation of the manuscript will profoundly increase the quality of the paper. Please find our comments for your review in the attachment.

Reviewer 2 Report
The article is a nice study and in general worth to be published. Unfortunately, the work has some weak points, which should be eliminated, if possible. Any improvements would be appreciated.
Line 102: The cationization approach is perhaps not optimal for this purpose, since all carrier proteins are severely changed in their properties. A less aggressive chemistry would be preferable.
The conjugation density should be determined preferentially by MALDI-TOF.
The affinity and cross-reactivity of the antibodies was not determined. In many cases, these properties are more important than the number of immunizations required.
Direct ELISA is often more informative to detect high-affinity antibodies.
I have difficulties to understand Table 1.
Inf Fig. 3, the concentrations of the inhibitors are not given. A full inhibition curve would be better.
Table 2 only shows the number of the hybridomas, but not their properties. Often one excellent clone is more valuable than 71 poor ones.
I could not find information about the used proteins, chemicals and other reagents or materials and devices.
Author Response

(The authors gave the same response as above.)

Round 2
Reviewer 2 Report
No further comments.